# Prevalence and risk factors of frailty in older adults with diabetes: A systematic review and meta-analysis

Yaqing Liu[1], Longhan Zhang[1], Xiaoyun Li[2], An Luo[1], Sixuan Guo[1], Xun Liu[1], Xingyu Wei[3], Yuanhong Sun[4], Manyi Wang[1], Li Liao[1]*

1 School of Nursing, University of South China, Hengyang, Hunan, China, 2 School of Computer, University of South China, Hengyang, Hunan, China, 3 Clinical Medical college of Acupuncture Moxibustion and Rehabilitation, Guangzhou University of Chinese Medicine, Guangzhou, Guangdong, China, 4 Hengyang Medical School, University of South China, Hengyang, Hunan, China

☯ These authors contributed equally to this work.

* liaoli@usc.edu.cn

**Data Availability Statement:** All relevant data are within the manuscript and its Supporting Information files.

**Funding:** This work was supported by the Natural Science Foundation of Hunan Province, China

## Abstract

### Objective

This systematic review and meta-analysis aimed to evaluate the prevalence of frailty and pre-frailty in older adults with diabetes; and to identify the risk factors associated with frailty in this population.

### Design

Systematic review and meta-analysis.

### Participants

24,332 people aged 60 years and older with diabetes.

### Methods

Six databases were searched (PubMed, Embase, the Cochrane Library, Web of Science, China Knowledge Resource Integrated Database, and Chinese Biomedical Database) up to 15 January 2024. Random effects models were used in instances of significant heterogeneity. Subgroup analysis and meta-regression were conducted to identify the potential source of heterogeneity. The Agency for Healthcare Research and Quality (AHRQ) and the Newcastle-Ottawa Scale (NOS) were applied to assess the quality of included studies.

### Results

3,195 abstracts were screened, and 39 full-text studies were included. In 39 studies with 24,332 older people with diabetes, the pooled prevalence of frailty among older adults with diabetes was 30.0% (95% CI: 23.6%-36.7%). Among the twenty-one studies involving 7,922 older people with diabetes, the pooled prevalence of pre-frailty was 45.1% (95% CI: 38.5%-51.8%). The following risk factors were associated with frailty among older adults

(grant number 2023JJ50137). The funder of this study had no role in study design, data collection, data analysis, data interpretation, or writing of the report.

**Competing interests:** The authors have declared that no competing interests exist.

with diabetes: older age (OR = 1.08, 95% CI: 1.04–1.13, $p<0.05$), high HbA1c (OR = 2.14, 95% CI: 1.30–3.50, $p<0.001$), and less exercise (OR = 3.11, 95% CI: 1.36–7.12, $p<0.001$).

## Conclusions

This suggests that clinical care providers should be vigilant in identifying frailty and risk factors of frailty while screening for and intervening in older adults with diabetes. However, there are not enough studies to identify comprehensive risk factors of frailty in older adults with diabetes.

## Trial registration

**PROSPERO registration number:** CRD42023470933.

## Introduction

Diabetes continues to be a significant public health challenge worldwide. According to the IDF Diabetes Atlas (10th edition), the estimated prevalence of diabetes among adults aged 75–79 will be 24.7% in 2045 [1]. With the world's population aging, the proportion of people over 60 with diabetes will continue to rise [2]. Evidence shows that frailty is becoming the third type of complication of diabetes in addition to the traditional microvascular and macrovascular complications [3]. Frailty is characterized by a decline in functioning across multiple physiological systems and an elevated vulnerability to stressors [4]. Frailty is a significant risk factor for death and disability in older adults with diabetes [5]. Moreover, the rate of frailty is 3–5 times higher in older adults with diabetes than in non-diabetics [6]. However, current estimates of the prevalence of frailty in older adults with diabetes vary substantially. Indeed, the reported prevalence of frailty among older adults with diabetes varies across the studies from 6.3% to 84.4% [7,8]. Several factors contribute to the significant variation in the prevalence of frailty, including: 1) differences in the characteristics of older adults across studies, such as the severity and duration of diabetes mellitus; 2) discrepancies in the assessment criteria used for frailty; and 3) variations in the clinical characteristics of the participants.

It is now clear that diabetes and frailty are closely intertwined and can interact in a vicious circle. Frailty determines the prognosis for older adults with diabetes [9]; moreover, scientific evidence suggests that both pre-frailty and frailty increase healthcare utilization and lead to adverse health outcomes in people with diabetes mellitus [10]. Indeed, frailty is not simply a consequence of aging and is not always progressive [11]; instead, the functional deficits linked to frailty in older adults are reversible and dynamic [12]. Therefore, understanding the true magnitude of frailty among patients with diabetes is imperative for implementing possible early screening and suitable intervention strategies, subsequently reducing the incidence of complications and improving the prognosis.

To date, no systematic review or meta-analysis has assessed the overall prevalence of frailty and pre-frailty in older adults with diabetes. Evidence from such a meta-analysis will provide robust information on the epidemiology of frailty and pre-frailty among older adults with diabetes, which would be necessary to plan early and suitable intervention strategies for those population groups. Therefore, this article aims to systematically analyze published studies on the prevalence of frailty and pre-frailty in older adults with diabetes and identify the risk factors.

## Methods

### Protocol and registration

This review has been registered in PROSPERO (registration number CRD42023470933). Searches of the International Prospective Register of Systematic Reviews (PROSPERO) have confirmed that no protocols for a meta-analysis with an identical scope exist.

### Design and selection criteria

We followed the methods recommended by the Cochrane Collaboration. We complied with the reporting standards of the 2020 Preferred Reporting Items for Systematic Review and Meta-analysis (PRISMA) guideline [13] (S1 Table). The PECOS (population, exposure, comparison, outcomes, study design) model was used to shape the clinical question and search strategy (S2 Table). The criteria for inclusion of the study: 1) people with diabetes aged ≥60 years; 2) exact diagnostic criteria for the frailty were available; 3) prevalence or risk factors of frailty were reported; 4) cross-sectional, case-control, or cohort study. The exclusion criteria: 1) incomplete data; 2) the language of the publication was other than English or Chinese; 3) the sample size was less than 100; 4) studies were assessed as high-risk of bias.

### Search strategy

We conducted a comprehensive literature search in the online databases of PubMed, Embase, the Cochrane Library, Web of Science, China Knowledge Resource Integrated Database (CNKI), and Chinese Biomedical Database (CBM) up to 15 January 2024 (S3 Table). Reference lists of all included studies and reviews were checked for possible other studies. Initial keywords included "Aged", "Elderly", "Frail Elderly", "old people", "elder*", "older people", "Diabetes Mellitus", "Diabetes Insipidus", "Diabetes Insipidus", "Glucose Intolerance", "Diabetes", "Glycuresis", "Diabetic", "type 2 diabetes", "Frailty", "Frailties", "Frailness", "frailty syndrome", "Debility", "Debilities", "prevalence*", "epidemiology", "incidence*", "morbidity", "factor*", "influencing factor*", "relevant factor*", "dangerous factor*", "factor*", "risk", "cohort studies", "case-control studies", "comparative study", "risk factors", "cohort", "compared", "groups", "case control", "multivariate". Each key search term's Medical Subject Heading (MeSH) and combinations were explored in every database. Boolean operators such as "AND" and "OR" were used to search the relevant studies.

### Study selection and data extraction

Upon execution of the searches, the complete list of records was stored and managed using the Rayyan platform (https://www.rayyan.ai) for screening [14]. The search, screening, and data extraction were performed independently by two reviewers (Liu and Zhang). Data extraction was accomplished through online document collaboration. Disagreements were resolved through discussion with the senior investigator (Li). The following information was recorded: demographic information (e.g., age, gender); methodological data included study design, sample size, data collection period, region, method(s) of diagnosis of frailty, and data collection method(s); outcome data included prevalence of frailty, prevalence of pre-frailty and risk factors reported. The missing 95% CI of OR will be calculated in logistic regression: $95\%\text{CI} = EXP(LOG(OR) \pm 1.96 \times SE)$.

### Quality assessment of studies

Two investigators (Liu and Zhang) evaluated the study quality based on the study designs. Conflicts were resolved by another reviewer (Luo). The 11-item criteria recommended by the

US Agency for Healthcare Quality and Research (AHRQ) was used to assess the risk of bias in cross-sectional studies. Studies scoring 0 to 3 are considered low quality, 4 to 7 indicate moderate quality, and 8 to 11 are regarded as high quality [15]. For cohort studies, the Newcastle-Ottawa Scale (NOS) was used. Three domains were evaluated with the following items: 1) selection, 2) comparability, and 3) exposure [16]. High quality was considered when the assigned score was≥8, moderate quality when the score was between four and seven, and low quality when the score was≤3. Studies with AHRQ and NOS scores of less than four were assessed as having a high risk of bias and were not included in the study.

### Data analysis

We used STATA 17.0 (Stata Corporation, College Station, TX) to perform the meta-analysis. A $p$-value of $<0.05$ was considered statistically significant. $I^2$ statistics and forest plots assessed heterogeneity. $I^2$ value between 25%-50% was regarded as low heterogeneity, 50%-75% moderate heterogeneity, and $I^2>75\%$ high heterogeneity [17]. A random effects model was used where moderate or high heterogeneity occurs; otherwise, the fixed effect model was used. Microsoft Excel was utilized to draw the forest plot based on the data. In general, there is a high degree of heterogeneity in prevalence studies; our analytic plan included an exploration of the causes of variation. We hypothesized that clinical and patient factors would be associated with the reported prevalence. Factors related to heterogeneity were identified via subgroup analyses in case of categorical factors (e.g., gender, regions, study design, study population, and tools of diagnosis of frailty) and meta-regression for continuous variables (e.g., year of publication, participants' mean age, and percentage of females). In the meta-regression results, $p<0.10$ was considered statistically significant due to the low power of these tests [18]. To assess the risk factors of frailty, the odds ratios (ORs) and associated 95% confidence intervals (CIs) were extracted from included studies. A funnel plot was used to assess publication bias [19], and asymmetry was tested using Egger's linear regression method ($p<0.05$ is considered significant) [20].

## Results

### Study process

The initial search yielded 4,209 results; after duplicates were removed, 3,195 records remained for screening. After the titles and abstract review, 96 articles were selected for full-text review to exclude those that were irrelevant or did not meet the pre-specified eligibility criteria. Finally, 39 studies (14 in Chinese and 25 in English) were utilized for the meta-analysis. The selection process is displayed within the PRISMA diagram (Fig 1). Of these articles, 39 studies reported frailty prevalence among older people with a confirmed diagnosis of diabetes, and 21 studies reported pre-frailty prevalence among older people with diabetes [21–41]. Additionally, we identified 13 studies on risk factors of frailty in older adults with diabetes [22,24,25,28–30,32,40,42–45], of which nine were included in the meta-analysis [22,24,25,28–30,32,40,43]. The reasons for excluding the 57 studies can be found in S4 Table.

### Study characteristics

The key characteristics of the included studies are presented in Table 1. These studies were conducted in Asia (n = 27, 69.2%) [21–35,38–40,42–50], Europe (n = 6, 15.4%) [7,36,37,51–53], North America (n = 5, 12.8%) [8,54–57] and South America (n = 1, 2.6%) [41]. Out of 39 articles, 29 were cross-sectional studies (9,371 patients) [8,21–35,37,38,40–43,45–47,49–52]; 10 were cohort studies [7,36,39,44,48,53–57] with 14,961 patients. Of the 39 included studies, 15

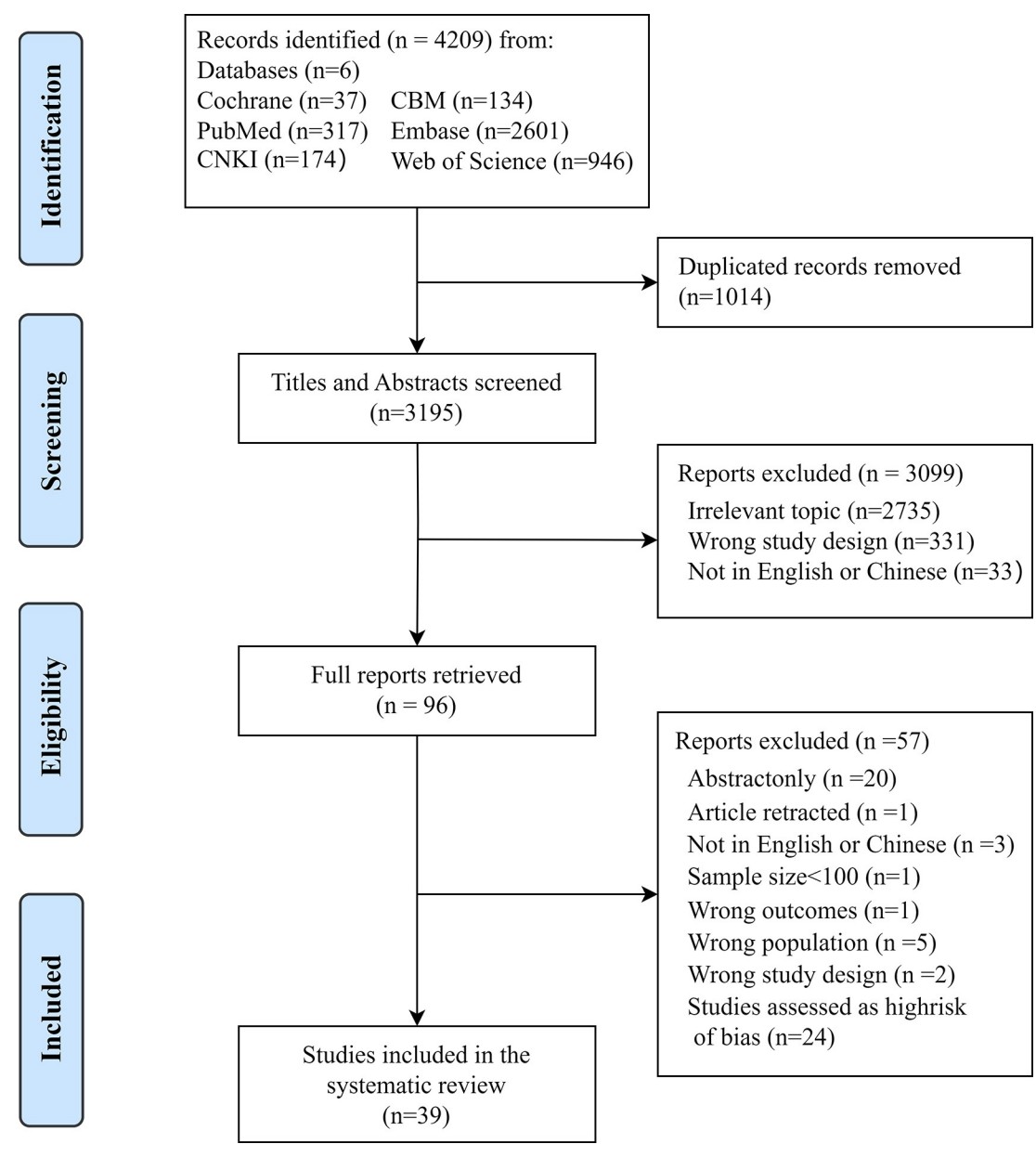

**Fig 1. Flow diagram of study selection.**

used the Frailty Phenotype to assess the frailty level of older people with diabetes [22,25,26,28,29,32,35,36,38–41,49,53,55]. Seven studies used the Frail scale [7,21,23,24,30,33,43], 5 used the Tilburg Frailty Indicator [42,45,46,50,51], 4 used the Clinical Frailty Scale [8,27,44,57], 4 used the Frailty Index [36,37,54,56]. One study used three different diagnostic criteria [36], and the rest used other diagnostic tools [31,34,47,48,52]. More detailed research data is available in S5 Table.

## Risk of bias assessment of included studies

The assessment of the risk of bias in the included studies using AHRQ for cross-sectional studies (n = 29) (S6 Table and S1 Fig). The final scores ranged between 4 and 7, indicating

**Table 1. Characteristics of included studies.** (n = 39).

| Study | Country | population | Study Design | Sample size | Age range | Female % | Frailty prevalence % | Pre-frailty prevalence % | Diagnostic criteria | Risk factors Assessed |
|---|---|---|---|---|---|---|---|---|---|---|
| Guo 2019 [21] | China | Hospitalized patients | cross-sectional study | 306 | 74.12 ±5.81 | 51.0% | 37.6% | 36.6% | Frail scale | - |
| Guo 2018 [22] | China | Hospitalized patients | cross-sectional study | 306 | 74.12 ±5.81 | 51.0% | 54.6% | 21.2% | FP | Age, nutritional status, HbA1c, CCI, Biguanides |
| Sun 2024 [46] | China | Hospitalized patients | cross-sectional study | 260 | ≥60 | 49.6% | 68.8% | - | TFI | - |
| Cheng 2020 [23] | China | Hospitalized patients | cross-sectional study | 998 | 66(63–70) | 51.3% | 8.0% | 55.3% | Frail scale | - |
| Ge X. 2020 [47] | China | Hospitalized patients | cross-sectional study | 221 | 69.85 ±4.68 | 54.8% | 62.0% | - | EFS | - |
| Jia 2019 [45] | China | Hospitalized patients | cross-sectional study | 296 | ≥65 | 42.2% | 46.3% | - | TFI | Comorbidity, SDSCA, HbA1c, Education, Polypharmacy, Smoking |
| Chen 2019 [24] | China | outpatient | cross-sectional study | 278 | 72.2 ±5.5 | 54.3% | 14.7% | 38.5% | Frail scale | Age, Exercise, HbA1c, Diabetes duration, Education, Per capita monthly household income (RMB) (ref:>2000), BMI, Number of chronic diseases |
| Wu 2021 [25] | China | Community-dwelling | cross-sectional study | 254 | ≥60 | 42.5% | 27.6% | 49.6% | FP | Alcohol, Drinking, Exercise, BMI, Per capita monthly household income (RMB)(ref:>5000) |
| Ge Q. 2020 [26] | China | Hospitalized patients | cross-sectional study | 137 | ≥60 | 52.6% | 16.8% | 45.3% | FP | - |
| Li 2022 [27] | China | Nursing Home | cross-sectional study | 351 | 72.0 ±7.7 | 47.9% | 27.4% | 12.5% | CFS | - |
| Deng 2020 [28] | China | Community-dwelling | cross-sectional study | 343 | ≥60 | 52.5% | 26.5% | 46.1% | FP | Depression |
| Zhao 2023 [29] | China | Hospitalized patients | cross-sectional study | 253 | 67 (62,74) | 47.8% | 33.6% | 38.7% | FP | Age, HbA1c, GLFS |
| Wang 2023 [30] | China | Hospitalized patients | cross-sectional study | 1652 | 74.5 ±6.42 | 46.5% | 26.5% | 59.7% | Frail scale | Age, Exercise, Education, Per capita monthly household income (RMB) (ref:>5000), Comorbidity, Depression |
| Hayakawa 2021 [31] | Japan | Community-dwelling | cross-sectional study | 149 | 72~81 | 45.0% | 9.4% | 52.3% | J-CHS | - |
| Kong 2021 [32] | China | Community-dwelling | cross-sectional study | 291 | 69(67–72) | 52.9% | 19.2% | 51.5% | FP | Alcohol Drinking, Malnutrition risk/ malnutrition, Exercise, Depression |
| Wang Q. 2023 [33] | China | outpatient | cross-sectional study | 168 | ≥65 | 51.8% | 22.6% | 49.4% | Frail scale | - |
| Nishimura 2019 [34] | Japan | outpatient | cross-sectional study | 213 | 70.4 ±5.6 | 49.3% | 18.3% | 38.0% | KCL | - |

*(Continued)*

**Table 1.** (Continued)

| Study | Country | population | Study Design | Sample size | Age range | Female % | Frailty prevalence % | Pre-frailty prevalence % | Diagnostic criteria | Risk factors Assessed |
|---|---|---|---|---|---|---|---|---|---|---|
| Kang 2021 [35] | Korea | Community-dwelling | cross-sectional study | 670 | 76.0 ±3.8 | 48.8% | 7.5% | 57.5% | FP | - |
| Qin 2023 [54] | USA | Community-dwelling | cohort study | 2894 | 65.43 (0.30) | 48.7% | 42.4% | - | FI | - |
| Zaslavsky 2016 [55] | USA | Community-dwelling | cohort study | 253 | ≥65 | - | 37.2% | - | FP | - |
| Yanagita 2018 [44] | Japan | Hospitalized patients | cohort study | 132 | 78.30 ±7.98 | 52.3% | 41.7% | - | CFS | Low HbA1c level |
| Lopez-Garcia 2018 [7] | Spain | female nurses | cohort study | 8970 | ≥60 | 100.0% | 6.3% | - | Frail scale | - |
| Sable-Morita 2021 [48] | Japan | outpatient | cohort study | 477 | 73.8 ±5.4 | 45.5% | 24.5% | - | KCL | - |
| Aguayo 2019 (1) [36] | UK | Community-dwelling | cohort study | 635 | 72 66, 77) | 46.0% | 52.9% | - | FI | - |
| Aguayo 2019 (2) [36] | UK | Community-dwelling | cohort study | 635 | 72 66, 77) | 46.0% | 18.9% | - | EFS | - |
| Aguayo 2019 (3) [36] | UK | Community-dwelling | cohort study | 635 | 72 66, 77) | 46.0% | 23.0% | 72.9% | FP | - |
| Ferri-Guerra 2020 [56] | USA | Community-dwelling | cohort study | 763 | 72.87 ±6.78 | 1.7% | 50.5% | - | FI | - |
| Muszalik 2022 [37] | Poland | outpatient | cross-sectional study | 103 | 72.96 ±7.55 | 63.1% | 25.2% | 31.1% | FI | - |
| Xiu 2020 [38] | China | Hospitalized patients | cross-sectional study | 240 | 68.89 ±6.88 | 48.3% | 15.0% | 45.8% | FP | Low FT3 |
| Bąk 2021 [51] | Poland | outpatient | cross-sectional study | 148 | ≥60 | 51.4% | 43.2% | - | TFI | - |
| Cacciatore 2013 [52] | Italy | Community-dwelling | cross-sectional study | 188 | 74.2 ±6.3 | 67.0% | 48.4% | - | FSS | - |
| Kitamura 2019 [39] | Japan | Community-dwelling | cohort study | 181 | 71.0 ±5.6 | 45.3% | 12.7% | 58.6% | FP | - |
| Hubbard 2010 [57] | Canada | Community-dwelling | cohort study | 310 | ≥70 | - | 42.3% | - | CFS | - |
| García-Esquinas 2015 [53] | Spain | Community-dwelling | cohort study | 346 | 69.4 ±6.4 | 41.0% | 11.3% | - | FP | - |
| MacKenzie 2020 [8] | Canada | Hospitalized patients | cross-sectional study | 141 | 80.6 ±7.8 | 52.5% | 84.4% | - | CFS | - |
| Nguyen 2020 [49] | Vietnam | Hospitalized patients | cross-sectional study | 176 | ≥60 | 60.2% | 18.8% | - | FP | - |
| Liu 2023 [43] | China | Hospitalized patients | cross-sectional study | 301 | 68 (64,74) | 44.9% | 23.6% | - | Frail scale | Age, Exercise, Hearing impairment, Fall history, Insulin therapy, Risk of malnutrition, Malnutrition, Depression |
| Sun 2020 [40] | China | Hospitalized patients | cross-sectional study | 269 | 68.9 ±6.8 | 50.2% | 15.6% | 44.2% | FP | Age, Female, eGFR, 25(OH)D3, Anaemia |

*(Continued)*

**Table 1.** (Continued)

| Study | Country | population | Study Design | Sample size | Age range | Female % | Frailty prevalence % | Pre-frailty prevalence % | Diagnostic criteria | Risk factors Assessed |
|---|---|---|---|---|---|---|---|---|---|---|
| Sun 2021 [50] | China | outpatient | cross-sectional study | 592 | ≥65 | 42.2% | 46.3% | - | TFI | - |
| Lin 2022 [42] | China | Community-dwelling | cross-sectional study | 248 | 73.9 ±5.9 | 49.6% | 26.6% | - | TFI | Age, Number of chronic diseases, ADL, Frequency of hyperglycemia, IADL ≥1 task disability, TGDS, MMSE Frequency of falling, CVA, Renal disease |
| Lima Filho 2020 [41] | Brazil | Hospitalized patients | cross-sectional study | 125 | 68.66 ±6.62 | - | 43.2% | 47.2% | FP | - |

Abbreviations: ADL: Activities of daily living; CCI: Charlson comorbidity index; CFS: Clinical Frail Scale; CVA: Cerebrovascular accident; EFS: Edmonton Frail Scale; eGFR: Estimated glomerular filtration rate; FI: Frailty Index; FP: Frailty Phenotype; FSS: Frailty staging system; FT3: Free triiodothyronine; GLFS: Geriatric locomotive function scale; HbA1c: glycated hemoglobin; IADL: Instrumental activities of daily living; J-CHS: Japanese version of the Cardiovascular Health Study; KCL: Kihon Checklist; MMSE: Mini-mental state examination; SDSCA: The Scale of Diabetes Self-Care Activities; TFI: Tilburg Frailty Indicator; TGDS: Taiwan geriatric depression scale; UK: United Kingdom; USA: United States of America.

moderate quality. The assessment of the risk of bias in the included studies using NOS for cohort studies. Among the 10 cohort studies, three studies (30%) were high-quality studies (scored 8), and the remaining were moderate-quality articles (scored between 4 and 7) (S7 Table and S2 Fig).

## Pooled prevalence of frailty prevalence in older adults with diabetes

The meta-analysis of the pooled prevalence of frailty prevalence in older adults with diabetes was 30.0% (95% CI: 23.6%-36.7%, $I^2$ = 99.2%, $p<0.001$) in 39 studies with 24,332 patients (S3 Fig). A summary of frailty prevalence is shown in Fig 2.

## Pooled prevalence of pre-frailty prevalence in older adults with diabetes

The meta-analysis of the pooled prevalence of pre-frailty prevalence in older adults with diabetes was 45.1% (95% CI: 38.5%-51.8%, $I^2$ = 97.1%, $p<0.001$) in 21 studies with 7,922 patients (S4 Fig). A summary of pre-frailty prevalence is shown in Fig 3.

## Risk factors of frailty in older adults with diabetes

The pooled analysis identified three potential risk factors associated with frailty in older adults with diabetes (Fig 4): older age (OR = 1.08, 95% CI: 1.04–1.13, $p<0.05$), high HbA1c (OR = 2.14, 95% CI: 1.30–3.50, $p<0.001$), less exercise (OR = 3.11, 95% CI: 1.36–7.12, $p<0.001$). The association of frailty with depression did not reach statistical significance (OR = 1.52, 95% CI: 1.25–1.85, $p = 0.16$). The meta-analysis only included risk factors reported in 3 or more articles to ensure the scientific validity of the results. Meta-analyses of other factors for frailty and risk factors for pre-frailty were not conducted due to insufficient information.

## Subgroup analysis and meta-regression

Subgroup analysis was done to identify the potential source of heterogeneity (Figs 2 and 3). Regarding gender, the prevalence of frailty among females was higher at 22.8% (95% CI:

**Summary of frailty prevalence**

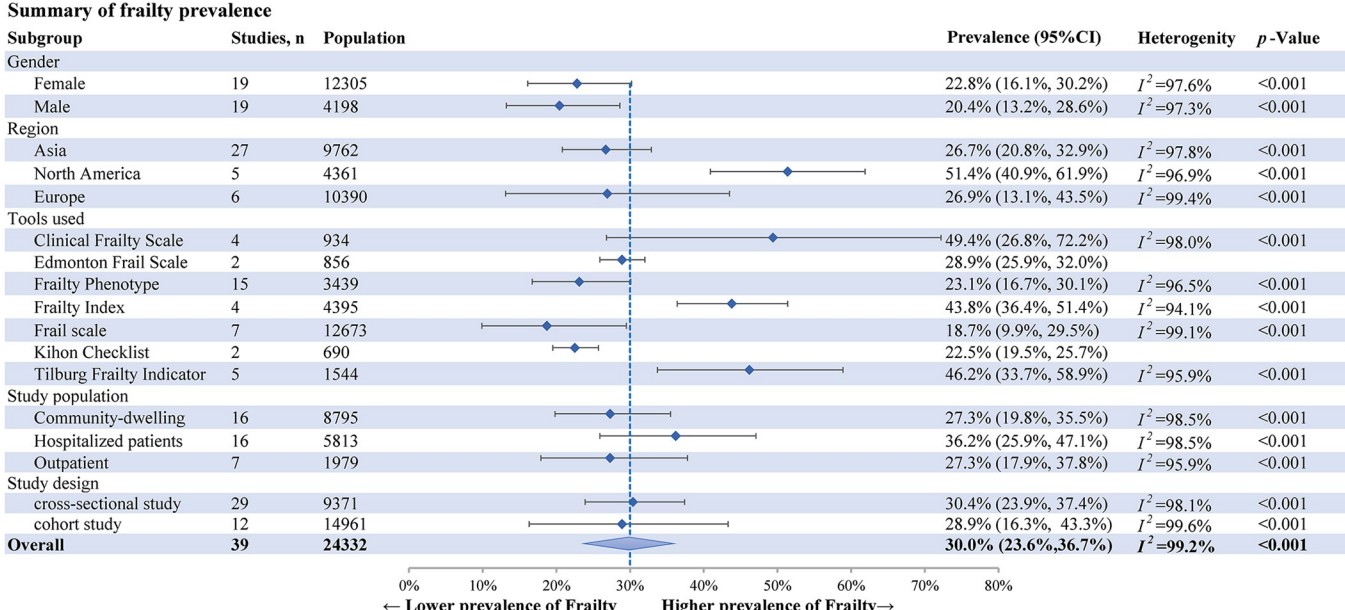

Fig 2. Summary of frailty prevalence.

16.1%-30.2%) than males at 20.4% (95% CI: 13.2%-28.6%). According to the grouping of the study population, the pooled prevalence of frailty was 27.3% among community residents, 36.2% among hospitalized patients, and 27.3% among outpatients; by region, the pooled prevalence of frailty was lowest in Asia (26.7%, 95% CI: 20.8%-32.9%), and highest in North America (51.4%, 95% CI: 40.9%-61.9%). Subgroup analysis was not conducted for risk factors due to insufficient data.

Meta-regression showed an inverse association between the prevalence of frailty and gender (female%). However, this inverse association was not statistically significant (regression = -0.133, 95% CI: (-0.409)—(0.502), $p > 0.1$). Also, an inverse association between the prevalence

**Summary of pre-frailty prevalence**

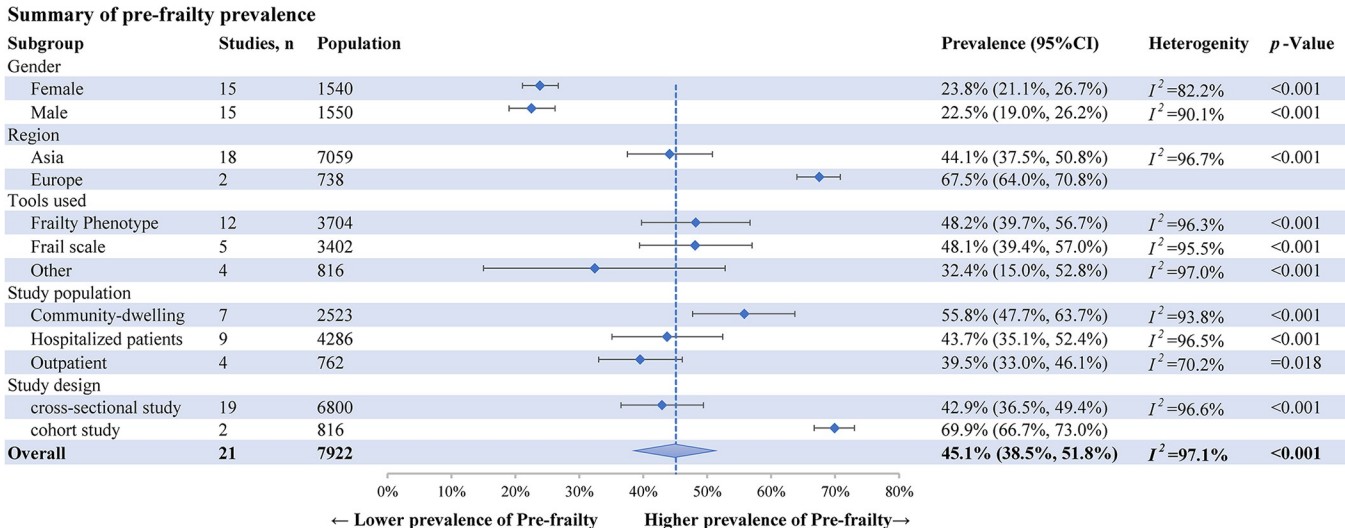

Fig 3. Summary of pre-frailty prevalence.

**Summary of the risk factors**

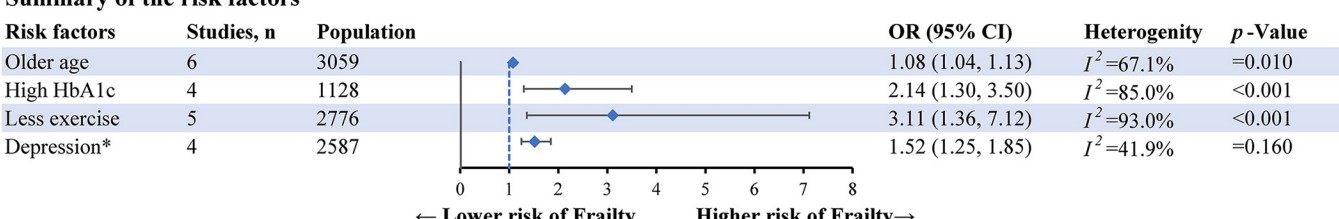

| Risk factors | Studies, n | Population | | OR (95% CI) | Heterogenity | p-Value |
|---|---|---|---|---|---|---|
| Older age | 6 | 3059 | | 1.08 (1.04, 1.13) | $I^2$ =67.1% | =0.010 |
| High HbA1c | 4 | 1128 | | 2.14 (1.30, 3.50) | $I^2$ =85.0% | <0.001 |
| Less exercise | 5 | 2776 | | 3.11 (1.36, 7.12) | $I^2$ =93.0% | <0.001 |
| Depression* | 4 | 2587 | | 1.52 (1.25, 1.85) | $I^2$ =41.9% | =0.160 |

**Fig 4. Summary of risk factors (*conducted a fixed-effects model).**

of Frailty and the year of publication was not statistically significant (regression = -0.007, 95% CI: (-0.038)—(0.025), $p$>0.1). But, there was a direct association between the prevalence of Frailty and mean age. This association was statistically significant (regression = 0.017, 95% CI: (-0.001)-(0.036), $p$<0.1). Moreover, gender (female%) (regression = 1.794, 95% CI: 1.106–2.483, $p$<0.001) were associated with prevalence of pre-frailty, whereas mean age (regression = -0.003, 95% CI: (-0.032)—(0.026), $p$>0.1) or year of publication (regression = 0.002, 95% CI: (-0.044)—(0.048), $p$>0.1) was not (S5 Fig).

## Publication bias

The funnel plot for frailty prevalence and pre-frailty prevalence asymmetry both revealed evidence of publication bias between study heterogeneity. Egger's test results further confirmed the funnel plots' asymmetry ($p$<0.05, see S6 and S7 Figs). Publication bias was not assessed for risk factors studies due to insufficient studies.

## Discussion

This study represents the first systematic review and meta-analysis examining the prevalence and risk factors of frailty among older adults with diabetes. Based on 39 studies included in the current meta-analysis, which involved a total of 24,332 older adults with diabetes, the pooled prevalence of frailty is 30.0%; the prevalence rate is significantly higher than that of the general elderly population in the United States (15.3%) [58], Japan (7.4%) [59] and China (9.9%) [60]. This may be attributed to patients with diabetes being in a hyperglycemic state, which impairs skeletal muscle regeneration, leading to reduced myocyte size and strength and thus accelerating the onset of frailty [61]. Frail older individuals are more susceptible to adverse events, including hypoglycemia and mortality [62]. Indeed, some defined models of frailty (e.g., Frailty Phenotype and Clinical Frailty Scale) have been used to set goals for diabetes medications in Europe and the United States [63,64]. Considering the high prevalence of frailty in older adults with diabetes, we suggest that clinicians should take frailty into more significant consideration when planning diabetes treatment for older adults. Our study also found that the prevalence of pre-frailty (45.1%) in older adults with diabetes was higher than the prevalence of frailty. This suggests that clinical care providers should be vigilant in identifying pre-frailty while screening for and intervening in frailty among older adults with diabetes [65].

### Impact of risk factors

The assessment of potential risk factors associated with frailty indicated a statistically significant correlation with three factors: older age, high HbA1c, and less exercise. In a previous study, these factors have also been demonstrated to correlate significantly with frailty among patients with diabetes [66]. However, there is controversy regarding the idea that HbA1c

influences the frailty condition of older adults with diabetes. Most studies suggest that a high level of HbA1c is a risk factor for frailty; still, a study conducted in Japan concluded that a low HbA1c level is an independent risk factor for frailty in older adults with type 2 diabetes mellitus [44]. One possible reason for this is the variation in the frailty scale used; among the studies that reported on HbA1c, three used the FP scale [22,29,32], one used the Frail scale [24], and the Japanese one used the Clinical Frail Scale [44]. Age is acknowledged as an independent risk factor for frailty, with its prevalence escalating as individuals grow older. This increase is attributed to age-related degeneration of organs and a reduction in reserve capacity [59]. The current study has demonstrated that older age is a risk factor for frailty in older adults with diabetes, consistent with previous findings [67].

Frailty is characterized by a disruption in the harmonious interplay among various domains, referred to as dimensions, which encompass biological, psychological, nutritional, and socioeconomic aspects [4]. Each of these dimensions plays a pivotal role in the complex etiology of frailty, and their interdependencies contribute to the overall decline in an individual's resilience and capacity to withstand stressors. Biologically, frailty may manifest as a decline in physiological reserve and homeostatic mechanisms, leading to a decreased ability to respond to and recover from stressors. This can include sarcopenia, the age-related loss of muscle mass and strength [68], as well as alterations in immune function and hormonal balance [69], which collectively impair the body's ability to maintain stability and resist disease. Psychologically, frailty is often accompanied by symptoms of depression, anxiety, and a general sense of helplessness. A recent bidirectional Mendelian randomization study demonstrated a bidirectional causal relationship between frailty and depression [70]. Additionally, frailty is linked to nutritional imbalances and deficiencies that can affect overall health and function. Protein-energy malnutrition, for example, is a common issue in frail individuals, leading to muscle wasting and reduced strength [71].

Moreover, frailty is relevant to socioeconomic factors such as lower income, educational level, and social support, which can limit access to healthcare resources and contribute to poor health status [24,25,45]. The lack of social support networks can lead to social isolation, a known risk factor for the development and progression of frailty [72]. Furthermore, detrimental lifestyle choices, such as smoking, excessive alcohol consumption, and inadequate personal hygiene, contribute to the heightened risk of developing frailty [73,74]. Given the multifaceted nature of frailty, investigations into its risk factors among elderly individuals with diabetes should encompass an examination of these diverse domains. However, due to the paucity of empirical data specifically about older adults with diabetes, the inclusion of these potential risk factors in the current study was not feasible.

### Subgroup analysis and meta-regression findings

The gender-stratified analysis showed that the prevalence of frailty in female patients was higher than in male patients (Fig 2), and it was in line with a previous report [75]. Furthermore, the results of meta-regression analyses also suggest that females are associated with a higher prevalence of pre-frailty (regression = 1.794, 95% CI: 1.106–2.483, $p < 0.001$, see S5 Fig). Current academic research indicates that women are susceptible to frailty, likely due to the rapid loss of estrogen in postmenopausal women. This hormonal change negatively affects muscular strength, neuromuscular function, and postural stability, consequently increasing the incidence of frailty in older women [76]. Additionally, some scholars have observed that men are more inclined to neglect and underreport their health issues compared to women. This tendency also contributes to a gender disparity in the prevalence of frailty [77].

The analysis of regional subgroups indicated that the prevalence of frailty is higher in North America than in Asia and Europe. This observation aligns with the findings of a previous

meta-analysis [65]. We discovered that in one of the North American studies, the participants had an average age of 80.6 years, and the prevalence of frailty was 84.4% [8]. To mitigate potential biases, we temporarily removed the paper with the high prevalence. However, even without that paper, the meta-analysis still indicated that the pooled prevalence of frailty in North America was 43.4% (95% CI: 38.4%-48.4%, p<0.001), which remained higher than that in Asia (26.7%) and Europe (26.9%). Further investigation into environmental influences and ethnic diversity's role in frailty may elucidate the causes of this regional variance.

The meta-analysis encompassed studies that used 11 different frailty assessment tools. The prevalence of frailty reported using the various assessment tools varied widely, ranging from 18.7% (Frailty Scale) to 49.4% (Clinical Frailty Scale). This variation is likely due to differences in patient populations and the sensitivity and specificity of these scales (Fig 2). One possible explanation for the higher prevalence found with the Clinical Frailty Scale is its primary use in assessing frailty in hospitalized patients. This group typically has a higher prevalence of frailty (36.2%) due to generally poorer health compared to community-dwelling individuals and outpatients. Given the worldwide variation in frailty assessment tools for older adults with diabetes, adopting a multi-disciplinary approach to standardize criteria is essential. Such standardization would contribute to improving clinical practice and ensuring more consistent assessment of frailty across different healthcare settings.

## Strengths and limitations of the study

The study presented multiple strengths. Firstly, it used a rigorous review methodology and delivered a comprehensive report. Additionally, it incorporated a large global sample that quantitatively synthesized the prevalence and risk factors of frailty among older adults with diabetes. Furthermore, subgroup analysis and meta-regression investigated the heterogeneous factors contributing to this demographic's estimated prevalence of frailty. This review offered the inaugural synthesis of evidence on the global prevalence of frailty in older individuals with diabetes, providing significant findings to clinicians, researchers, and policymakers. The information provided could assist clinicians in optimizing care for older individuals with diabetes by customizing treatment objectives and protocols according to their frailty status. This personalized approach was crucial for enhancing patient outcomes and quality of life.

The meta-analysis exhibited limitations, notably the high heterogeneity ($I^2$ = 41.9%–99.6%) among the included studies. Despite subgroup analyses, this heterogeneity remained pronounced, likely due to the extensive scope of the systematic review and the substantial volume of data involved. Our analysis was based on data from published studies. Due to this limitation, we could not discount other potential risk factors at that time, including the impact of poor lifestyles like smoking and alcohol abuse. Most studies included in our analysis did not specify the diabetes typology. Consequently, our study did not distinguish between the different types of diabetes in people. In addition, the study's scope was constrained by the inclusion of only English and Chinese language studies, which may have affected its comprehensiveness.

## Conclusion

In conclusion, this systematic review revealed a pooled frailty prevalence of 30.0% and a prefrailty prevalence of 45.1% among older individuals with diabetes. These high rates underscore the urgency of increasing public awareness about frailty and pre-frailty. While current literature may not suffice to draw comprehensive conclusions beyond pinpointing older age, high HbA1c, and less exercise as risk factors, this review summarizes the available evidence and catalyzes further high-quality research in this field.

## Supporting information

**S1 Fig. Critical appraisal of the 27 cross-sectional studies.**
(PDF)

**S2 Fig. Critical appraisal of the 11 cohort studies.**
(PDF)

**S3 Fig. Pooled prevalence of frailty.**
(PDF)

**S4 Fig. Pooled prevalence of pre-frailty.**
(PDF)

**S5 Fig. Meta regression analysis.**
(PDF)

**S6 Fig. Publication bias assessment of frailty prevalence.**
(PDF)

**S7 Fig. Publication bias assessment of pre-frailty prevalence.**
(PDF)

**S1 Table. PRISMA 2020 checklist.**
(PDF)

**S2 Table. PECOS model.**
(DOCX)

**S3 Table. The search and screening strategy.**
(DOCX)

**S4 Table. Excluded studies and reason (N = 57).**
(DOCX)

**S5 Table. Detailed data of studies included.**
(XLSX)

**S6 Table. Assessment of the quality of cross-sectional studies based on the criteria recommended by the US Agency for Healthcare Quality and Research (AHRQ).**
(DOCX)

**S7 Table. Quality of cohort studies based on the Newcastle–Ottawa scale.**
(DOCX)

## Author Contributions

**Data curation:** Yaqing Liu, An Luo.

**Formal analysis:** Yaqing Liu, Longhan Zhang, Yuanhong Sun.

**Funding acquisition:** Li Liao.

**Methodology:** Longhan Zhang, An Luo, Xingyu Wei.

**Project administration:** Li Liao.

**Resources:** Yaqing Liu, Xun Liu, Li Liao.

**Software:** Longhan Zhang, Xiaoyun Li, Xun Liu, Yuanhong Sun.

**Supervision:** Longhan Zhang, Xiaoyun Li, Xun Liu, Manyi Wang, Li Liao.

**Visualization:** Sixuan Guo.

**Writing – original draft:** Yaqing Liu, Longhan Zhang, Xingyu Wei.

**Writing – review & editing:** Yaqing Liu, Longhan Zhang, Xiaoyun Li, An Luo, Sixuan Guo, Xingyu Wei, Manyi Wang, Li Liao.

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
