## [Decision Letter · Decision Letter 0]

6 Aug 2024

PONE-D-24-25621Prevalence and risk factors of frailty in older adults with diabetes: A systematic review and meta-analysisPLOS ONE

Dear Dr. LIAO,

Thank you for submitting your manuscript to PLOS ONE. After careful consideration, we feel that it has merit but does not fully meet PLOS ONE’s publication criteria as it currently stands. Therefore, we invite you to submit a revised version of the manuscript that addresses the points raised during the review process.

We look forward to receiving your revised manuscript.

Kind regards,

Francesco Curcio, M.D., Ph.D.

Academic Editor

PLOS ONE

https://bmcgeriatr.biomedcentral.com/articles/10.1186/s12877-020-01830-2

https://pubmed.ncbi.nlm.nih.gov/33906698/

In your revision ensure you cite all your sources (including your own works), and quote or rephrase any duplicated text outside the methods section. Further consideration is dependent on these concerns being addressed.

3. Please amend your authorship list in your manuscript file to include author Yuanhong Sun.

Additional Editor Comments:

According to Reviewers' revision the manuscript need a minor revision.

Reviewers' comments:

Reviewer's Responses to Questions

**Comments to the Author**

1. Is the manuscript technically sound, and do the data support the conclusions?

Reviewer #1: Yes

Reviewer #2: Yes

2. Has the statistical analysis been performed appropriately and rigorously? 

Reviewer #1: Yes

Reviewer #2: Yes

3. Have the authors made all data underlying the findings in their manuscript fully available?

Reviewer #1: Yes

Reviewer #2: Yes

4. Is the manuscript presented in an intelligible fashion and written in standard English?

Reviewer #1: Yes

Reviewer #2: Yes

5. Review Comments to the Author

Reviewer #1: The manuscript is interesting and topic. I have onòly a main concern. The frailty is actually considered a multidimensional frailty and 4 different domains should be considered (physical, mental, nutritional and social). Please discuss this matter in the revised form of the manuscript.

Reviewer #2: This study represents the first systematic review and meta-analysis examining the prevalence and risk factors of frailty among older adults with diabetes. I found the study of interest and methodologically correct. The data derived from 39 studies including 24,332 people aged 60 years and older with diabetes. The pooled prevalence of frailty among older adults with diabetes was 30.0%. (95% CI: 23.6%-36.7%). Among the twenty-one studies involving 7,922 older people with diabetes, the pooled prevalence of pre-frailty was 45.1% (95% CI: 38.5%-51.8%). The following risk factors were associated with frailty among older adults with diabetes: older age (OR=1.08, 95% CI: 1.04-1.13, p<0.05), high HbA1c (OR=2.14, 95% CI: 1.30-3.50, p<0.001), and less exercise (OR=3.11, 95% CI: 1.36-7.12, p<0.001).

The study is well conducted. Data suffer from the different instrument used in literature to identify frailty status. But the analysis was conducted in order to analyze these differences.

6. PLOS authors have the option to publish the peer review history of their article (what does this mean?). If published, this will include your full peer review and any attached files.

Reviewer #1: No

Reviewer #2: **Yes: **francesco cacciatore

---

## [Author Response · Author response to Decision Letter 0]

8 Aug 2024

Dear Editor and Reviewers:

Thank you for your decision and constructive comments on our manuscript. We have carefully considered the suggestions of the editor and reviewers and made several changes. We have made our best efforts to improve the manuscript. The comments are laid out below in italicized font and specific concerns have been numbered. Our response is given in normal font and changes/additions to the manuscript are given in the yellow text.

Academic Editor:

1)Please ensure that your manuscript meets PLOS ONE's style requirements, including those for file naming.

Thanks for your reminder. We have modified the reference style of the manuscript according to PLOS ONE's style requirements. In addition, we have added the title of the figures.

2)We noticed you have some minor occurrence of overlapping text with the following previous publication(s)

We appreciate your thorough review. We have revised the overlapping text in the manuscript. (P2 Line26-28, P3 Line67-71, P3 Line82-83 and P18 Line53-54 marked in yellow in the revised paper)

3)Please amend your authorship list in your manuscript file to include author Yuanhong Sun.

Thanks for your reminder. The author Yuanhong Sun will be added to the list of authors via the submission system. (P1 Line4 marked in yellow in the revised paper)

4)Please include captions for your Supporting Information files at the end of your manuscript, and update any in-text citations to match accordingly.

We appreciate your attention to detail. We have added the captions of the supporting information and cited them in the corresponding location in the body of the text.

5)Please review your reference list to ensure that it is complete and correct. 

The reference list has been checked and modified.

Reviewer#1: 

The frailty is actually considered a multidimensional frailty and 4 different domains should be considered (physical, mental, nutritional and social). Please discuss this matter in the revised form of the manuscript.

We sincerely appreciate your valuable comments. We have addressed this issue in the revised manuscript. (P18 Line63-90 marked in yellow in the revised paper)

Reviewer#2:

Data suffer from the different instrument used in literature to identify frailty status. But the analysis was conducted in order to analyze these differences.

We sincerely appreciate the valuable comments. We provide a detailed discussion of these differences through subgroup analyses and meta-regression. (P20 Line114-125 marked in yellow in the revised paper)

Thanks again for your professional review work on our article. If you have any queries, please don’t hesitate to contact me at the address below.

Yours sincerely,

Li Liao

School of Nursing, University of South China, Hengyang, Hunan Province, China

liaoli@usc.edu.cn

---

## [Decision Letter · Decision Letter 1]

20 Aug 2024

Prevalence and risk factors of frailty in older adults with diabetes: A systematic review and meta-analysis

PONE-D-24-25621R1

Dear Dr. Liao,

We’re pleased to inform you that your manuscript has been judged scientifically suitable for publication and will be formally accepted for publication once it meets all outstanding technical requirements.

Kind regards,

Jacopo Sabbatinelli, MD, PhD

Academic Editor

PLOS ONE

Additional Editor Comments (optional):

Reviewers' comments:

Reviewer's Responses to Questions

**Comments to the Author**

1. If the authors have adequately addressed your comments raised in a previous round of review and you feel that this manuscript is now acceptable for publication, you may indicate that here to bypass the “Comments to the Author” section, enter your conflict of interest statement in the “Confidential to Editor” section, and submit your "Accept" recommendation.

Reviewer #1: All comments have been addressed

Reviewer #2: All comments have been addressed

2. Is the manuscript technically sound, and do the data support the conclusions?

Reviewer #1: Yes

Reviewer #2: Yes

3. Has the statistical analysis been performed appropriately and rigorously? 

Reviewer #1: Yes

Reviewer #2: Yes

4. Have the authors made all data underlying the findings in their manuscript fully available?

Reviewer #1: Yes

Reviewer #2: Yes

5. Is the manuscript presented in an intelligible fashion and written in standard English?

Reviewer #1: Yes

Reviewer #2: Yes

6. Review Comments to the Author

Reviewer #1: The manuscript is really impoeved and it merits to be published in PONE. No furtther comments are requested.

Reviewer #2: The manuscript is improved and I am of the opinion that the paper is of interest and is appropriate for publication.

7. PLOS authors have the option to publish the peer review history of their article (what does this mean?). If published, this will include your full peer review and any attached files.

Reviewer #1: No

Reviewer #2: **Yes: **francesco cacciatore

---

## [Editor Report · Acceptance letter]

29 Aug 2024

PONE-D-24-25621R1 

PLOS ONE

Dear Dr. Liao, 

I'm pleased to inform you that your manuscript has been deemed suitable for publication in PLOS ONE. Congratulations! Your manuscript is now being handed over to our production team.

Kind regards, 

on behalf of

Dr. Jacopo Sabbatinelli 

Academic Editor

PLOS ONE